# Analyzing the Harmonic Structure
# in Graph-Based Learning

**Xiao-Ming Wu**[1], **Zhenguo Li**[3], and **Shih-Fu Chang**[1,2]

[1]Department of Electrical Engineering, Columbia University
[2]Department of Computer Science, Columbia University
[3]Huawei Noah's Ark Lab, Hong Kong
{xmwu, sfchang}@ee.columbia.edu,   li.zhenguo@huawei.com

## Abstract

We find that various well-known graph-based models exhibit a common important *harmonic* structure in its target function – the value of a vertex is approximately the weighted average of the values of its adjacent neighbors. Understanding of such structure and analysis of the loss defined over such structure help reveal important properties of the target function over a graph. In this paper, we show that the variation of the target function across a cut can be upper and lower bounded by the ratio of its harmonic loss and the cut cost. We use this to develop an analytical tool and analyze five popular graph-based models: absorbing random walks, partially absorbing random walks, hitting times, pseudo-inverse of the graph Laplacian, and eigenvectors of the Laplacian matrices. Our analysis sheds new insights into several open questions related to these models, and provides theoretical justifications and guidelines for their practical use. Simulations on synthetic and real datasets confirm the potential of the proposed theory and tool.

## 1   Introduction

Various graph-based models, regardless of application, aim to learn a target function on graphs that well respects the graph topology. This has been done under different motivations such as Laplacian regularization [4, 5, 6, 14, 24, 25, 26], random walks [17, 19, 23, 26], hitting and commute times [10], p-resistance distances [1], pseudo-inverse of the graph Laplacian [10], eigenvectors of the Laplacian matrices [18, 20], diffusion maps [8], to name a few. Whether these models can capture the graph structure faithfully, or whether their target functions possess desirable properties over the graph, remain unclear. Understanding of such issues can be of great value in practice and has attracted much attention recently [16, 22, 23].

Several important observations about learning on graphs have been reported. Nadler et al. [16] showed that the target functions of Laplacian regularized methods become flat as the number of unlabeled points increases, but they also observed that a good classification can still be obtained if an appropriate threshold is used. An explanation to this would be interesting. Von Luxburg et al. [22] proved that commute and hitting times are dominated by the local structures in large graphs, ignoring the global patterns. Does this mean these metrics are flawed? Interestingly, despite this finding, the pseudo-inverse of graph Laplacian, known as the kernel matrix of commute times, consistently performs superior in collaborative filtering [10]. In spectral clustering, the eigenvectors of the normalized graph Laplacian are more desired than those of the un-normalized one [20, 21]. Also for the recently proposed partially absorbing random walks [23], certain setting of absorption rates seems better than others. While these issues arise from seemingly unrelated contexts, we will show in this paper that they can be addressed in a single framework.

Our starting point is the discovery of a common structure hidden in the target functions of various graph models. That is, the value of a vertex is approximately the weighted average of the values of its adjacent neighbors. We call this structure *the harmonic structure* for its resemblance to the harmonic function [9, 26]. It naturally arises from the first step analysis of random walk models, and, as will be shown in this paper, implicitly exists in other methods such as pseudo-inverse of the graph Laplacian and eigenvectors of the Laplacian matrices. The target functions of these models are characterized by their harmonic loss, a quantitative notion introduced in this paper to measure the discrepancy of a target function $f$ on cuts of graphs. The variations of $f$ across cuts can then be upper and lower bounded by the ratio of its harmonic loss and the cut cost. As long as the harmonic loss varies slowly, the graph conductance dominates the variations of $f$ – it will remain smooth in a dense area but vary sharply otherwise. Models possessing such properties successfully capture the cluster structures, and as shown in Sec. 4, lead to superior performance in practical applications including classification and retrieval.

This novel perspective allows us to give a unified treatment of graph-based models. We use this tool to study five popular models: absorbing random walks, partially absorbing random walks, hitting times, pseudo-inverse of the graph Laplacian, and eigenvectors of the Laplacian matrices. Our analysis provides new theoretical understandings into these models, answers related open questions, and helps to correct and justify their practical use. The key message conveyed in our results is that various existing models enjoying the harmonic structure are actually capable of capturing the global graph topology, and understanding of this structure can guide us in applying them properly.

## 2 Analysis

Let us first define some notations. In this paper, we consider graphs which are connected, undirected, weighted, and without self-loops. Denote by $\mathcal{G} = (\mathcal{V}, W)$ a graph with $n$ vertices $\mathcal{V}$ and a symmetric non-negative affinity matrix $W = [w_{ij}] \in \mathbb{R}^{n \times n}$ ($w_{ii} = 0$). Denote by $d_i = \sum_j w_{ij}$ the degree of vertex $i$, by $D = \text{diag}(d_1, d_2, \ldots, d_n)$ the degree matrix, and by $L = D - W$ the graph Laplacian [7]. The conductance of a subset $\mathcal{S} \subset \mathcal{V}$ of vertices is defined as $\Phi(\mathcal{S}) = \frac{w(\mathcal{S}, \bar{\mathcal{S}})}{\min(d(\mathcal{S}), d(\bar{\mathcal{S}}))}$, where $w(\mathcal{S}, \bar{\mathcal{S}}) = \sum_{i \in \mathcal{S}, j \in \bar{\mathcal{S}}} w_{ij}$ is the cut cost between $\mathcal{S}$ and its complement $\bar{\mathcal{S}}$, and $d(\mathcal{S}) = \sum_{i \in \mathcal{S}} d_i$ is the volume of $\mathcal{S}$. For any $i \notin \mathcal{S}$, denote by $i \sim \mathcal{S}$ if there is an edge between vertex $i$ and the set $\mathcal{S}$.

**Definition 2.1 (Harmonic loss).** *The harmonic loss of $f : \mathcal{V} \to \mathbb{R}$ on any $\mathcal{S} \subseteq \mathcal{V}$ is defined as:*

$$\mathcal{L}_f(\mathcal{S}) := \sum_{i \in \mathcal{S}} d_i \left( f(i) - \sum_{j \sim i} \frac{w_{ij}}{d_i} f(j) \right) = \sum_{i \in \mathcal{S}} \left( d_i f(i) - \sum_{j \sim i} w_{ij} f(j) \right). \tag{1}$$

Note that $\mathcal{L}_f(\mathcal{S}) = \sum_{i \in \mathcal{S}} (Lf)(i)$. By definition, the harmonic loss can be negative. However, as we shall see below, it is always non-negative on superlevel sets.

The following lemma shows that the harmonic loss couples the cut cost and the discrepancy of the function across the cut. This observation will serve as the foundation of our analysis in this paper.

**Lemma 2.2.** $\mathcal{L}_f(\mathcal{S}) = \sum_{i \in \mathcal{S}, j \in \bar{\mathcal{S}}} w_{ij}(f(i) - f(j))$. *In particular, $\mathcal{L}_f(\mathcal{V}) = 0$.*

In practice, to examine the variation of $f$ on a graph, one does not necessarily examine on every subset of vertices, which will be exponential in the number of vertices. Instead, it suffices to consider its variation on the superlevel sets defined as follows.

**Definition 2.3 (Superlevel set).** *For any function $f : \mathcal{V} \to \mathbb{R}$ on a graph and a scalar $c \in \mathbb{R}$, the set $\{i \mid f(i) \geq c\}$ is called a superlevel set of $f$ with level $c$.*

W.l.o.g., we assume the vertices are sorted such that $f(1) \geq f(2) \geq \cdots \geq f(n-1) \geq f(n)$. The subset $\mathcal{S}_i := \{1, \ldots, i\}$ is the superlevel set with level $f(i)$ if $f(i) > f(i+1)$. For convenience, we still call $\mathcal{S}_i$ a superlevel set of $f$ even if $f(i) = f(i+1)$. In this paper, we will mainly examine the variation of $f$ on its $n$ superlevel sets $\mathcal{S}_1, \ldots, \mathcal{S}_n$. Our first observation is that the harmonic loss on each superlevel set is non-negative, stated as follows.

**Lemma 2.4.** $\mathcal{L}_f(\mathcal{S}_i) \geq 0, i = 1, \ldots, n$.

Based on the notion of superlevel sets, it becomes legitimate to talk about the continuity of a function on graphs, which we formally define as follows.

**Definition 2.5** (**Continuity**). *For any function $f : \mathcal{V} \to \mathbb{R}$, we call it left-continuous if $i \sim \mathcal{S}_{i-1}$, $i = 2, \ldots, n$; we call it right-continuous if $i \sim \bar{\mathcal{S}}_i$, $i = 1, \ldots, n-1$; we call it continuous if $i \sim \mathcal{S}_{i-1}$ and $i \sim \bar{\mathcal{S}}_i$, $i = 2, \ldots, n-1$. Particularly, $f$ is called left-continuous, right-continuous, or continuous at vertex $i$ if $i \sim \mathcal{S}_{i-1}$, $i \sim \bar{\mathcal{S}}_i$, or $i \sim \mathcal{S}_{i-1}$ and $i \sim \bar{\mathcal{S}}_i$, respectively.*

**Proposition 2.6.** *For any function $f : \mathcal{V} \to \mathbb{R}$ and any vertex $1 < i < n$, 1) if $\mathcal{L}_f(i) < 0$, then $i \sim \mathcal{S}_{i-1}$, i.e., $f$ is left-continuous at $i$; 2) if $\mathcal{L}_f(i) > 0$, then $i \sim \bar{\mathcal{S}}_i$, i.e., $f$ is right-continuous at $i$; 3) if $\mathcal{L}_f(i) = 0$ and $f(i-1) > f(i) > f(i+1)$, then $i \sim \mathcal{S}_{i-1}$ and $i \sim \bar{\mathcal{S}}_i$, i.e., $f$ is continuous at $i$.*

The variation of $f$ can be characterized by the following upper and lower bounds.

**Theorem 2.7** (**Dropping upper bound**). *For $i = 1, \ldots, n-1$,*

$$f(i) - f(i+1) \leq \frac{\mathcal{L}_f(\mathcal{S}_i)}{w(\mathcal{S}_i, \bar{\mathcal{S}}_i)} = \frac{\mathcal{L}_f(\mathcal{S}_i)}{\Phi(\mathcal{S}_i) \min(d(\mathcal{S}_i), d(\bar{\mathcal{S}}_i))}. \tag{2}$$

**Theorem 2.8** (**Dropping lower bound**). *For $i = 1, \ldots, n-1$,*

$$f(u) - f(v) \geq \frac{\mathcal{L}_f(\mathcal{S}_i)}{w(\mathcal{S}_i, \bar{\mathcal{S}}_i)} = \frac{\mathcal{L}_f(\mathcal{S}_i)}{\Phi(\mathcal{S}_i) \min(d(\mathcal{S}_i), d(\bar{\mathcal{S}}_i))}, \tag{3}$$

*where $u := \arg \max\limits_{j \in \mathcal{S}_i, j \sim \bar{\mathcal{S}}_i} f(j)$ and $v := \arg \min\limits_{j \in \bar{\mathcal{S}}_i, j \sim \mathcal{S}_i} f(j)$.*

The key observations are two-fold. First, for any function $f$ on a graph, as long as its harmonic loss $\mathcal{L}_f(\mathcal{S}_i)$ varies slowly on the superlevel sets, i.e., $f$ is *harmonic almost everywhere*, the graph conductance $\Phi(\mathcal{S}_i)$ will dominate the variation of $f$. In particular, by Theorem 2.7, $f(i+1)$ drops little if $\Phi(\mathcal{S}_i)$ is large, whereas by Theorem 2.8, a big gap exists across the cut if $\Phi(\mathcal{S}_i)$ is small (see Sec. 3.1 for illustration). Second, the continuity (either left, right, or both) of $f$ ensures that its variations conform with the graph connectivity, i.e., points with similar values on $f$ tend to be connected. It is a desired property because a "discontinuous" function that changes alternatively among different clusters can hardly describe the graph. These observations can guide us in identifying "good" functions that encode the global structure of graphs, as will be shown in the next section.

## 3 Examples

With the tool developed in Sec. 2, in this section, we study five popular graph models arising from different contexts including SSL, retrieval, recommendation, and clustering. For each model, we show its target function in harmonic forms, quantify its harmonic loss, analyze its dropping bounds, and provide corrections or justifications for its use.

### 3.1 Absorbing Random Walks

The first model we examine is the seminal Laplacian regularization method [26] proposed for SSL. While it has a nice interpretation in terms of absorbing random walks, with the labeled points being absorbing states, it was argued in [16] that this method might be ill-posed for large unlabeled data in high dimension ($\geq 2$) because the target function is extremely flat and thus seems problematic for classification. [1] further connected this argument with the resistance distance on graphs, pointing out that the classification biases to the labeled points with larger degrees. Here we show that Laplacian regularization can actually capture the global graph structure and a simple normalization scheme would resolve the raised issue.

For simplicity, we consider the binary classification setting with one label in each class. Denote by $f : \mathcal{V} \to \mathbb{R}$ the absorption probability vector from every point to the positive labeled point. Assume the vertices are sorted such that $1 = f(1) > f(2) \geq \cdots \geq f(n-1) > f(n) = 0$ (vertex 1 is labeled positive and vertex $n$ is labeled negative). By the first step analysis of the random walk,

$$f(i) = \sum_{k \sim i} \frac{w_{ik}}{d_i} f(k), \ \text{for } i = 2, \ldots, n-1. \tag{4}$$

Our first observation is that the harmonic loss of $f$ is constant w.r.t. $\mathcal{S}_i$, as shown below.

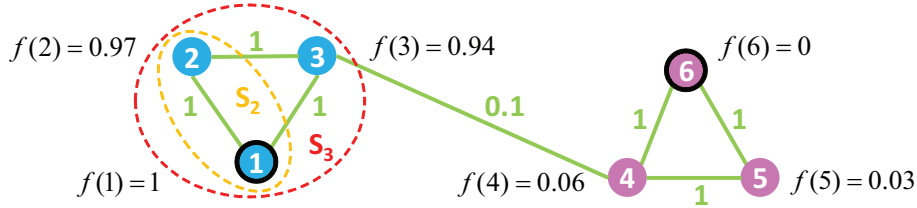

Figure 1: Absorbing random walks on a 6-point graph.

**Corollary 3.1.** $\mathcal{L}_f(\mathcal{S}_i) = \sum_{k \sim 1} w_{1k}(1 - f(k))$, $i = 1, \dots, n-1$.

The following statement shows that $f$ changes continuously on graphs under general condition.

**Corollary 3.2.** *Suppose $f$ is mutually different on unlabeled data. Then $f$ is continuous.*

Since the harmonic loss of $f$ is a constant on the superlevel sets $\mathcal{S}_i$ (Corollary 3.1), by Theorems 2.7 and 2.8, the variation of $f$ depends solely on the cut value $w(\mathcal{S}_i, \bar{\mathcal{S}}_i)$, which indicates that it will drop slowly when the cut is dense but drastically when the cut is sparse. Also by Corollary 3.2, $f$ is continuous. Therefore, we conclude that $f$ is a good function on graphs.

This can be illustrated by a toy example in Fig. 1, where the graph consists of 6 points in 2 classes denoted by different colors, with 3 points in each. The edge weights are all 1 except for the edge between the two cluster, which is 0.1. Vertices 1 and 6 (black edged) are labeled. The absorption probabilities from all the vertices to vertex 1 are computed and shown. We can see that since the cut $w(\mathcal{S}_2, \bar{\mathcal{S}}_2) = 2$ is quite dense, the drop between $f(2)$ and $f(3)$ is upper bounded by a small number (Theorem 2.7), so $f(3)$ must be very close to $f(2)$, as observed. In contrast, since the cut $w(\mathcal{S}_3, \bar{\mathcal{S}}_3) = 0.1$ is very weak, Theorem 2.8 guarantees that there will be a huge gap between $f(3)$ and $f(4)$, as also verified. The bound in Theorem 2.8 is now tight as there is only 1 edge in the cut.

Now let $f_1$ and $f_2$ denote the absorption probability vectors to the two labeled points respectively. To classify an unlabeled point $i$, the usual way is to compare $f_1(i)$ and $f_2(i)$, which is equivalent to setting the threshold as 0 in $f_0 = f_1 - f_2$. It was observed in [16] that although $f_0$ can be extremely flat in the presence of large unlabeled data in high dimension, setting the "right" threshold can produce sensible results. Our analysis explains this – it is because both $f_1$ and $f_2$ are informative of the cluster structures. Our key argument is that Laplacian regularization actually carries sufficient information about the graph structure, but how to exploit it can really make a difference.

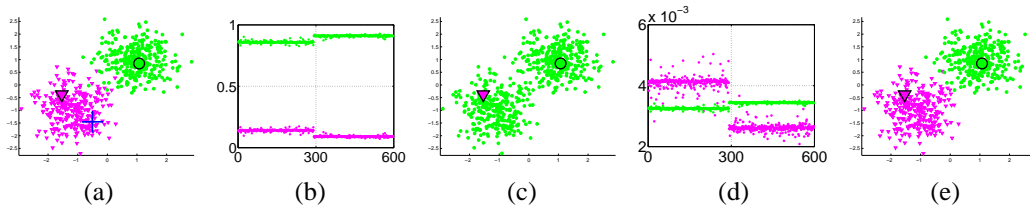

Figure 2: (a) Two 20-dimensional Gaussians with the first two dimensions plotted. The magenta triangle and the green circle denote labeled data. The blue cross denotes a starting vertex indexed by $i$ for later use. (b) Absorption probabilities to the two labeled points. (c) Classification by comparing the absorption probabilities. (d) Normalized absorption probabilities. (e) Classification by comparing the normalized absorption probabilities.

We illustrate this point by using a mixture of two 20-dimensional Gaussians of 600 points, with one label in each Gaussian (Fig. 2(a)). The absorption probabilities to both labeled points are shown in Fig. 2(b), in magenta and green respectively. The green vector is well above the the magenta vector, indicating that every unlabeled point has larger absorption probability to the green labeled point. Comparing them classifies all the unlabeled points to the green Gaussian (Fig. 2(c)). Since the green labeled point has larger degree than the magenta one[1], this result is expected from the analysis in [1]. However, the probability vectors are informative, with a clear gap between the clusters in each

vector. To use the information, we propose to normalize each vector by its probability mass, i.e., $f'(i) = f(i)/\sum_j f(j)$ (Fig. 2(d)). Comparing them leads to a perfect classification (Fig. 2(e)).

This idea is based on two observations from our analysis: 1) the variance of the probabilities within each cluster is small; 2) there is a gap between the clusters. The small variance indicates that comparing the probabilities is essentially the same as comparing their means within clusters. The gap between the clusters ensures that the normalization makes the vectors align well (this point is made precise in Supplement). Our above analysis applies to multi-class problems and allows more than one labeled points in one class. In this general case, the classification rule is as follows: 1) compute the absorption probability vector $f_i : \mathcal{U} \to \mathbb{R}$ for each labeled point $i$ by taking all other labeled points as negative, where $\mathcal{U}$ denotes the set of unlabeled points; 2) normalize $f_i$ by its mass, denoted by $f'_i$; 3) assign each unlabeled point $j$ to the class of $j^* := \arg\max_i\{f'_i(j)\}$. We denote this algorithm as ARW-N-1NN.

## 3.2 Partially Absorbing Random Walks

Here we revisit the recently proposed partially absorbing random walks (PARW) [23], which generalizes absorbing random walks by allowing partial absorption at each state. The absorption rate $p_{ii}$ at state $i$ is defined as $p_{ii} = \frac{\alpha\lambda_i}{\alpha\lambda_i + d_i}$, where $\alpha > 0$, $\lambda_i > 0$ are regularization parameters. Given current state $i$, a PARW in the next step will get absorbed at $i$ with probability $p_{ii}$ and with probability $(1 - p_{ii}) \times \frac{w_{ij}}{d_i}$ moves to state $j$. Let $a_{ij}$ be the probability that a PARW starting from state $i$ gets absorbed at state $j$ within finite steps, and denote by $A = [a_{ij}] \in \mathbb{R}^{n \times n}$ the absorption probability matrix. Then $A = (\alpha\Lambda + L)^{-1}\alpha\Lambda$, where $\Lambda = \mathrm{diag}(\lambda_1, \ldots, \lambda_n)$ is the regularization matrix.

PARW is a unified framework with several popular SSL methods and PageRank [17] as its special cases, corresponding to different $\Lambda$. Particularly, the case $\Lambda = I$ has been justified in capturing the cluster structures [23]. In what follows, we extend this result to show that the columns of $A$ obtained by PARW with almost arbitrary $\Lambda$ (not just $\Lambda = I$) actually exhibit strong harmonic structures and should be expected to work equally well.

Our first observation is that while $A$ is not symmetric for arbitrary $\Lambda$, $A\Lambda^{-1} = (\alpha\Lambda + L)^{-1}\alpha$ is.

**Lemma 3.3.** $a_{ij} = \frac{\lambda_j}{\lambda_i}a_{ji}$.

**Lemma 3.4.** $a_{ii}$ is the only largest entry in the $i$-th column of $A$, $i = 1, \ldots, n$.

Our second observation is that the harmonic structure exists in the probabilities of PARW from every vertex getting absorbed at a particular vertex, i.e., in the columns of $A$. W.l.o.g., consider the first column of $A$ and denote it by $p$. Assume that the vertices are sorted such that $p(1) > p(2) \geq \cdots \geq p(n-1) \geq p(n)$, where $p(1) > p(2)$ is due to Lemma 3.4. By the first step analysis of PARW, we can write $p$ in a recursive form:

$$p(1) = \frac{\alpha\lambda_1}{d_1 + \alpha\lambda_1} + \sum_{k \sim 1}\frac{w_{1k}}{d_1 + \alpha\lambda_1}p(k), \quad p(i) = \sum_{k \sim i}\frac{w_{ik}}{d_i + \alpha\lambda_i}p(k), \; i = 2, \ldots, n, \quad (5)$$

which is equivalent to the following harmonic form:

$$p(1) = \frac{\alpha\lambda_1}{d_1}(1 - p(1)) + \sum_{k \sim 1}\frac{w_{1k}}{d_1}p(k), \quad p(i) = -\frac{\alpha\lambda_i}{d_i}p(i) + \sum_{k \sim i}\frac{w_{ik}}{d_i}p(k), \; i = 2, \ldots, n. \quad (6)$$

The harmonic loss of $p$ can be computed from Eq. (6).

**Corollary 3.5.** $\mathcal{L}_p(\mathcal{S}_i) = \alpha\lambda_1(1 - \sum_{k \in \mathcal{S}_i} a_{1k}) = \alpha\lambda_1\sum_{k \in \bar{\mathcal{S}}_i} a_{1k}$, $i = 1, \ldots, n-1$.

**Corollary 3.6.** $p$ is left-continuous.

Now we are ready to examine the variation of $p$. Note that $\sum_k a_{1k} = 1$ and $a_{1k} \to \lambda_k/\sum_i \lambda_i$ as $\alpha \to 0$ [23]. By Theorem 2.7, the drop of $p(i)$ is upper bounded by $\alpha\lambda_1/w(\mathcal{S}_i, \bar{\mathcal{S}}_i)$, which is small when the cut $w(\mathcal{S}_i, \bar{\mathcal{S}}_i)$ is dense and $\alpha$ is small. Now let $k$ be the largest number such that $d(\mathcal{S}_k) \leq \frac{1}{2}d(\mathcal{V})$, and assume $\sum_{i \in \bar{\mathcal{S}}_k} \lambda_i \geq \frac{1}{2}\sum_i \lambda_i$. By Theorem 2.8, for $1 \leq i \leq k$, the drop of $p(i)$ across the cut $\{\mathcal{S}_i, \bar{\mathcal{S}}_i\}$ is lower bounded by $\frac{1}{3}\alpha\lambda_1/w(\mathcal{S}_i, \bar{\mathcal{S}}_i)$, if $\alpha$ is sufficiently small. This shows that $p(i)$ will drop a lot when the cut $w(\mathcal{S}_i, \bar{\mathcal{S}}_i)$ is weak. The comparison between the corresponding row and column of $A$ is shown in Figs. 3(a–b)[2], which confirms our analysis.

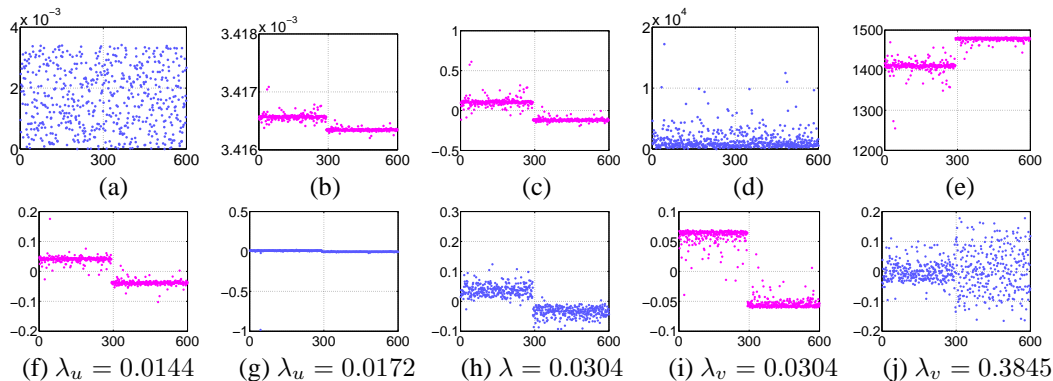

(f) $\lambda_u = 0.0144$  (g) $\lambda_u = 0.0172$  (h) $\lambda = 0.0304$  (i) $\lambda_v = 0.0304$  (j) $\lambda_v = 0.3845$

Figure 3: (a) Absorption probabilities that a PAWR gets absorbed at other points when starting from $i$ (see Fig. 2). (b) Absorption probabilities that PAWR gets absorbed at $i$ when starting from other points. (c) The $i$-th row of $L^\dagger$. (d) Hitting times from $i$ to hit other points. (e) Hitting times from other points to hit $i$. (f) and (g) Eigenvectors of $L$ ($\min_i\{d_i\} = 0.0173$). (h) An eigenvector of $L_{sym}$. (i) and (j) Eigenvectors of $L_{rw}$. The values in (f–j) denote eigenvalues.

It is worth mentioning that our analysis substantially extends the results in [23] by showing that the setting of $\Lambda$ is not really necessary – a random $\Lambda$ can perform equally well if using the columns instead of the rows of $A$. In addition, our result includes the seminal local clustering model [2] as a special case, which corresponds to $\Lambda = D$ in our analysis.

## 3.3 Pseudo-inverse of the Graph Laplacian

The pseudo-inverse $L^\dagger$ of the graph Laplacian is a valid kernel corresponding to commute times [10, 12]. While commute times may fail to capture the global topology in large graphs [22], $L^\dagger$, if used directly as a similarity measure, gives superior performance in practice [10]. Here we provide a formal analysis and justification for $L^\dagger$ by revealing the strong harmonic structure hidden in it.

**Lemma 3.7.** $(L^\dagger L)_{ij} = -\frac{1}{n}$, $i \neq j$; and $(L^\dagger L)_{ii} = 1 - \frac{1}{n}$.

Note that $L^\dagger$ is symmetric since $L$ is symmetric. W.l.o.g., we consider the first row of $L^\dagger$ and denote it by $\ell$. The following lemma shows the harmonic form of $\ell$.

**Lemma 3.8.** $\ell$ has the following harmonic form:

$$\ell(1) = \frac{1 - \frac{1}{n}}{d_1} + \sum_{k \sim 1} \frac{w_{1k}}{d_1} \ell(k), \ \ell(i) = -\frac{\frac{1}{n}}{d_i} + \sum_{k \sim i} \frac{w_{ik}}{d_i} \ell(k), \ i = 2, \ldots, n. \tag{7}$$

W.l.o.g., assume the vertices have been sorted such that $\ell(1) > \ell(2) \geq \cdots \geq \ell(n-1) \geq \ell(n)$[3]. Then the harmonic loss of $\ell$ on the set $\mathcal{S}_i$ admits a very simple form, as shown below.

**Corollary 3.9.** $\mathcal{L}_\ell(\mathcal{S}_i) = \frac{|\bar{\mathcal{S}}_i|}{n}$, $i = 1, \ldots, n-1$.

**Corollary 3.10.** $\ell$ is left-continuous.

By Corollary 3.9, $\mathcal{L}_\ell(\mathcal{S}_i) < 1$ and decreases very slowly in large graphs since $\mathcal{L}_\ell(\mathcal{S}_i) - \mathcal{L}_\ell(\mathcal{S}_{i+1}) = \frac{1}{n}$ for any $i$. From the analysis in Sec. 2, we can immediately conclude that the variation of $\ell(i)$ is dominated by the cut cost on the superlevel set $\mathcal{S}_i$. Fig. 3(c) illustrates this argument.

## 3.4 Hitting Times

The hitting time $h_{ij}$ from vertex $i$ to $j$ is the expected number of steps it takes a random walk starting from $i$ to reach $j$ for the first time. While it was proven in [22] that hitting times are dominated by the local structure of the target, we show below that the hitting times from other points to the same target admit a harmonic structure, and thus are still able to capture the global structure of graphs. Our result is complementary to the analysis in [22], and provides a justification of using hitting times in information retrieval where the query is taken as the target to be hit by others [15].

Let $h : \mathcal{V} \to \mathbb{R}$ be the hitting times from every vertex to a particular vertex. W.l.o.g., assume the vertices have been sorted such that $h(1) \geq h(2) \geq \cdots \geq h(n-1) > h(n) = 0$, where vertex $n$ is the target vertex. Applying the first step analysis, we obtain the harmonic form of $h$:

$$h(i) = 1 + \sum_{k \sim i} \frac{w_{ik}}{d_i} h(k), \quad \text{for } i = 1, \ldots, n-1. \tag{8}$$

The harmonic loss on the set $\mathcal{S}_i$ turns out to be the volume of the set, as stated below.

**Corollary 3.11.** $\mathcal{L}_h(\mathcal{S}_i) = \sum\limits_{1 \leq k \leq i} d_k, \, i = 1, \ldots, n-1.$

**Corollary 3.12.** $h$ *is right-continuous.*

Now let us examine the variation of $h$ across any cut $\{\mathcal{S}_i, \bar{\mathcal{S}}_i\}$. Note that

$$\frac{\mathcal{L}_h(\mathcal{S}_i)}{w(\mathcal{S}_i, \bar{\mathcal{S}}_i)} = \frac{\alpha_i}{\Phi(\mathcal{S}_i)}, \text{ where } \alpha_i = \frac{d(\mathcal{S}_i)}{\min(d(\mathcal{S}_i), d(\bar{\mathcal{S}}_i))}. \tag{9}$$

First, by Theorem 2.8, there could be a significant gap between the target and its neighbors, since $\alpha_{n-1} = \frac{d(\mathcal{V})}{d_n} - 1$ could be quite large. As $i$ decreases from $d(\mathcal{S}_i) > \frac{1}{2} d(\mathcal{V})$, the variation of $\alpha_i$ becomes slower and slower ($\alpha_i = 1$ when $d(\mathcal{S}_i) \leq \frac{1}{2} d(\mathcal{V})$), so the variation of $h$ will depend on the variation of the conductance of $\mathcal{S}_i$, i.e., $\Phi(\mathcal{S}_i)$, according to Theorems 2.7 and 2.8. Fig. 3(e) shows that $h$ is flat within the clusters, but there is a large gap presented between them. In contrast, there are no gaps exhibited in the hitting times from the target to other vertices (Fig. 3(d)).

### 3.5 Eigenvectors of the Laplacian Matrices

The eigenvectors of the Laplacian matrices play a key role in graph partitioning [20]. In practice, the eigenvectors with smaller (positive) eigenvalues are more desired than those with larger eigenvalues, and the ones from a normalized Laplacian are preferred than those from the un-normalized one. These choices are usually justified from the relaxation of the normalized cuts [18] and ratio cuts [11]. However, it has been known that these relaxations can be arbitrarily loose [20]. It seems more interesting if one can draw conclusions by analyzing the eigenvectors directly. Here we address these issues by examining the harmonic structures in these eigenvectors.

We follow the notations in [20] to denote two normalized graph Laplacians: $L_{rw} := D^{-1}L$ and $L_{sym} := D^{-\frac{1}{2}}LD^{-\frac{1}{2}}$. Denote by $u$ and $v$ two eigenvectors of $L$ and $L_{rw}$ with eigenvalues $\lambda_u > 0$ and $\lambda_v > 0$, respectively, i.e., $Lu = \lambda_u u$ and $L_{rw}v = \lambda_v v$. Then we have

$$u(i) = \sum_{k \sim i} \frac{w_{ik}}{d_i - \lambda_u} u(k), \quad v(i) = \sum_{k \sim i} \frac{w_{ik}}{d_i(1 - \lambda_v)} v(k), \quad \text{for } i = 1, \ldots, n. \tag{10}$$

We can see that the smaller $\lambda_u$ and $\lambda_v$, the stronger the harmonic structures of $u$ and $v$. This explains why in practice the eigenvector with the second[4] smallest eigenvalues gives superior performance. As long as $\lambda_u \ll \min_i \{d_i\}$, we are safe to say that $u$ will have a significant harmonic structure, and thus will be informative for clustering. However, if $\lambda_u$ is close to $\min_i \{d_i\}$, no matter how small $\lambda_u$ is, the harmonic structure of $u$ will be weaker, and thus $u$ is less useful. In contrast, from Eq. (10), $v$ will always enjoy a significant harmonic structure as long as $\lambda_v$ is much smaller than 1. This explains why eigenvectors of $L_{rw}$ are preferred than those of $L$ for clustering. These arguments are validated in Figs. 3(f–j), where we also include an eigenvector of $L_{sym}$ for comparison.

## 4 Experiments

In the first experiment[5], we test absorbing random walks (ARW) for SSL, with the class mass normalization suggested in [26] (ARW-CMN), our proposed normalization (ARW-N-1NN, Sec. 3.1), and without any normalization (ARW-1NN) – where each unlabeled instance is assigned the class of the labeled instance at which it most likely gets absorbed. We also compare with the local and global

Table 1: Classification accuracy on 9 datasets.

| | USPS | YaleB | satimage | imageseg | ionosphere | iris | protein | spiral | soybean |
|---|---|---|---|---|---|---|---|---|---|
| ARW-N-1NN | **.879** | **.892** | **.777** | **.673** | **.771** | **.918** | **.589** | **.830** | **.916** |
| ARW-1NN | .445 | .733 | .650 | .595 | .699 | .902 | .440 | .754 | .889 |
| ARW-CMN | .775 | .847 | .741 | .624 | .724 | .894 | .511 | .726 | .856 |
| LGC | .821 | .884 | .725 | .638 | .731 | .903 | .477 | .729 | .816 |
| PARW ($\Lambda = I$) | .880 | .906 | .781 | .665 | .752 | .928 | .572 | .835 | .905 |

consistency (LGC) method [24] and the PARW with $\Lambda = I$ in [23]. The results are summarized in Table 1. We can see that ARW-N-1NN and PARW ($\Lambda = I$) consistently perform the best, which verifies our analysis in Sec. 3. The results of ARW-1NN are unsatisfactory due to its bias to the labeled instance with the largest degree [1]. Although ARW-CMN does improve over ARW-1NN in many cases, it does not perform as well as ARW-N-1NN, mainly because of the artifacts induced by estimating the class proportion from limited labeled data. The results of LGC are not comparable to ARW-N-1NN and PARW ($\Lambda = I$), which is probably due to the lack of a harmonic structure.

Table 2: Ranking results (MAP) on USPS.

| Digits | 0 | 1 | 2 | 3 | 4 | 5 | 6 | 7 | 8 | 9 | All |
|---|---|---|---|---|---|---|---|---|---|---|---|
| $\Lambda = R$ (column) | **.981** | **.988** | **.875** | **.892** | **.647** | **.780** | **.941** | **.918** | **.746** | **.731** | **.850** |
| $\Lambda = R$ (row) | .169 | .143 | .114 | .096 | .092 | .076 | .093 | .093 | .075 | .086 | .103 |
| $\Lambda = I$ | .981 | .988 | .876 | .893 | .646 | .778 | .940 | .919 | .746 | .730 | .850 |

In the second experiment, we test PARW on a retrieval task on USPS (see Supplement). We compare the cases with $\Lambda = I$ and $\Lambda = R$, where $R$ is a random diagonal matrix with positive diagonal entries. For $\Lambda = R$, we also compare the uses of columns and rows for retrieval. The results are shown in Table 2. We observe that the columns in $\Lambda = R$ give significantly better results compared with rows, implying that the harmonic structure is vital to the performance. $\Lambda = R$ (column) and $\Lambda = I$ perform very similarly. This suggests that it is not the special setting of absorbing rates but the harmonic structure that determines the overall performance.

Table 3: Classification accuracy on USPS.

| $k$-NN unweighted graphs | 10 | 20 | 50 | 100 | 200 | 500 |
|---|---|---|---|---|---|---|
| HT($\mathcal{L} \to \mathcal{U}$) | **.8514** | **.8361** | **.7822** | **.7500** | **.7071** | **.6429** |
| HT($\mathcal{U} \to \mathcal{L}$) | .1518 | .1454 | .1372 | .1209 | .1131 | .1113 |
| $L^{\dagger}$ | .8512 | .8359 | .7816 | .7493 | .7062 | .6426 |

In the third experiment, we test hitting times and pseudo-inverse of the graph Laplacian for SSL on USPS. We compare two different uses of hitting times, the case of starting from the labeled data $\mathcal{L}$ to hit the unlabeled data $\mathcal{U}$ (HT($\mathcal{L} \to \mathcal{U}$)), and the case of the opposite direction (HT($\mathcal{U} \to \mathcal{L}$)). Each unlabeled instance $j$ is assigned the class of labeled instance $j^*$, where $j^* = \arg\min_{i \in \mathcal{L}}\{h_{ij}\}$ in HT($\mathcal{L} \to \mathcal{U}$), $j^* = \arg\min_{i \in \mathcal{L}}\{h_{ji}\}$ in (HT($\mathcal{U} \to \mathcal{L}$)), and $j^* = \arg\max_{i \in \mathcal{L}}\{\ell_{ji}\}$ in $L^{\dagger} = (\ell_{ij})$. The results averaged over 100 trials are shown in Table 3, where we see that HT($\mathcal{L} \to \mathcal{U}$) performs much better than HT($\mathcal{U} \to \mathcal{L}$), which is expected as the former admits a desired harmonic structure. Note that HT($\mathcal{L} \to \mathcal{U}$) is not lost as the number of neighbors increases (i.e., the graph becomes more connected). The slight performance drop is due to the inclusion of more noisy edges. In contrast, HT($\mathcal{U} \to \mathcal{L}$) is completely lost [20]. We also observe that $L^{\dagger}$ produces very competitive performance, which again supports our analysis.

## 5   Conclusion

In this paper, we explore *the harmonic structure* that widely exists in graph models. Different from previous research [3, 13] of harmonic analysis on graphs, where the selection of canonical basis on graphs and the asymptotic convergence on manifolds are studied, here we examine how functions on graphs deviate from being harmonic and develop bounds to analyze their theoretical behavior. The proposed harmonic loss quantifies the discrepancy of a function across cuts, allows a unified treatment of various models from different contexts, and makes them easy to analyze. Due to its resemblance with standard mathematical concepts such as divergence and total variation, an interesting line of future work is to make their connections clear. Other future works include deriving more rigorous bounds for certain functions and extending our analysis to more graph models.

## Footnotes

[1] The degrees are 1.4405 and 0.1435. We use a weighted 20-NN graph (see Supplement).

[2] $\lambda_i$'s are sampled from the uniform distribution on the interval $[0, 1]$ and $\alpha = 1e-6$, as used in Sec. 4.

[3] $\ell(1) > \ell(2)$ since one can show that any diagonal entry in $L^\dagger$ is the only largest in the corresponding row.

[4]Note that the smallest one is zero in either $L$ or $L_{rw}$.

[5]Please see Supplement for parameter settings, data description, graph construction, and experimental setup.

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
