[Supplementary Material · nips2013_sup.pdf]

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

where in the last equation we use the fact that $\sum_{i \in \mathcal{S}, j \in \mathcal{S}} w_{ij}(f(i) - f(j)) = 0$, due to the symmetry of the affinity ($w_{ij} = w_{ij}$). $\quad\blacksquare$

In practice, to examine the variation of $f$ on a graph, one does not necessarily examine on every subset of vertices, which will be exponential in the number of vertices. Instead, it suffices to consider its variation on the superlevel sets defined as follows.

**Definition 2.3 (Superlevel set).** *For any function $f : \mathcal{V} \to \mathbb{R}$ on a graph and a scalar $c \in \mathbb{R}$, the set $\{i \mid f(i) \geq c\}$ is called a superlevel set of $f$ with level $c$.*

W.l.o.g., we assume the vertices are sorted such that $f(1) \geq f(2) \geq \cdots \geq f(n-1) \geq f(n)$. The subset $\mathcal{S}_i := \{1, \ldots, i\}$ is the superlevel set with level $f(i)$ if $f(i) > f(i+1)$. For convenience, we still call $\mathcal{S}_i$ a superlevel set of $f$ even if $f(i) = f(i+1)$. In this paper, we will mainly examine the variation of $f$ on its $n$ superlevel sets $\mathcal{S}_1, \ldots, \mathcal{S}_n$. Our first observation is that the harmonic loss on each superlevel set is non-negative, stated as follows.

**Lemma 2.4.** $\mathcal{L}_f(\mathcal{S}_i) \geq 0, i = 1, \ldots, n.$

*Proof.* This follows from Lemma 2.2 and the fact that $f(k) \geq f(j), \forall k \in \mathcal{S}_i$ and $\forall j \in \bar{\mathcal{S}}_i$. $\quad\blacksquare$

Based on the notion of superlevel sets, it becomes legitimate to talk about the continuity of a function on graphs, which we formally define as follows.

**Definition 2.5 (Continuity).** *For any function $f : \mathcal{V} \to \mathbb{R}$, we call it left-continuous if $i \sim \mathcal{S}_{i-1}$, $i = 2, \ldots, n$; we call it right-continuous if $i \sim \bar{\mathcal{S}}_i$, $i = 1, \ldots, n-1$; we call it continuous if $i \sim \mathcal{S}_{i-1}$ and $i \sim \bar{\mathcal{S}}_i$, $i = 2, \ldots, n-1$. Particularly, $f$ is called left-continuous, right-continuous, or continuous at vertex $i$ if $i \sim \mathcal{S}_{i-1}$, $i \sim \bar{\mathcal{S}}_i$, or $i \sim \mathcal{S}_{i-1}$ and $i \sim \bar{\mathcal{S}}_i$, respectively.*

**Proposition 2.6.** *For any function $f : \mathcal{V} \to \mathbb{R}$ and any vertex $1 < i < n$, 1) if $\mathcal{L}_f(i) < 0$, then $i \sim \mathcal{S}_{i-1}$, i.e., $f$ is left-continuous at $i$; 2) if $\mathcal{L}_f(i) > 0$, then $i \sim \bar{\mathcal{S}}_i$, i.e., $f$ is right-continuous at $i$; 3) if $\mathcal{L}_f(i) = 0$ and $f(i-1) > f(i) > f(i+1)$, then $i \sim \mathcal{S}_{i-1}$ and $i \sim \bar{\mathcal{S}}_i$, i.e., $f$ is continuous at $i$.*

*Proof.* 1) By Definition 2.1, if $\mathcal{L}_f(i) < 0$, we have $f(i) - \sum_{j \sim i} \frac{w_{ij}}{d_i} f(j) < 0$. Then there must exist some $j \sim i$, such that $f(j) > f(i)$, implying $j \in \mathcal{S}_{i-1}$ and thus $i \sim \mathcal{S}_{i-1}$. Result 2) can be shown similarly.

3) As we assume the graph is connected, we must have $i \sim \mathcal{S}_{i-1}$ or $i \sim \bar{\mathcal{S}}_i$. Suppose $i \sim \mathcal{S}_{i-1}$. Then there exists some $j \in \mathcal{S}_{i-1}$ such that $f(j) > f(i)$ since $f(i-1) > f(i)$. Since $\mathcal{L}_f(i) = 0$, i.e., $f(i) - \sum_{j \sim i} \frac{w_{ij}}{d_i} f(j) = 0$, there must exist some $k \sim i$ such that $f(k) < f(i)$, i.e., $i \sim \bar{\mathcal{S}}_i$. Similarly, we can show that if $i \sim \bar{\mathcal{S}}_i$ then $i \sim \mathcal{S}_{i-1}$. $\quad\blacksquare$

The variation of $f$ can be characterized by the following upper and lower bounds.

**Theorem 2.7 (Dropping upper bound).** *For $i = 1, \ldots, n-1$,*

$$f(i) - f(i+1) \leq \frac{\mathcal{L}_f(\mathcal{S}_i)}{w(\mathcal{S}_i, \bar{\mathcal{S}}_i)} = \frac{\mathcal{L}_f(\mathcal{S}_i)}{\Phi(\mathcal{S}_i) \min(d(\mathcal{S}_i), d(\bar{\mathcal{S}}_i))}. \tag{2}$$

*Proof.* By Lemma 2.2, we have

$$\mathcal{L}_f(\mathcal{S}_i) = \sum_{j \in \mathcal{S}_i, k \in \bar{\mathcal{S}}_i} w_{jk}(f(j) - f(k)) \geq \sum_{j \in \mathcal{S}_i, j \in \bar{\mathcal{S}}_i} w_{jk}(f(i) - f(i+1))$$

$$\geq (f(i) - f(i+1)) \sum_{j \in \mathcal{S}_i, k \in \bar{\mathcal{S}}_i} w_{jk} = (f(i) - f(i+1))w(\mathcal{S}_i, \bar{\mathcal{S}}_i),$$

which concludes the proof. $\quad\blacksquare$

**Theorem 2.8 (Dropping lower bound).** *For $i = 1, \ldots, n-1$,*

$$f(u) - f(v) \geq \frac{\mathcal{L}_f(\mathcal{S}_i)}{w(\mathcal{S}_i, \bar{\mathcal{S}}_i)} = \frac{\mathcal{L}_f(\mathcal{S}_i)}{\Phi(\mathcal{S}_i) \min(d(\mathcal{S}_i), d(\bar{\mathcal{S}}_i))}, \tag{3}$$

*where $u := \arg \max_{j \in \mathcal{S}_i, j \sim \bar{\mathcal{S}}_i} f(j)$ and $v := \arg \min_{j \in \bar{\mathcal{S}}_i, j \sim \mathcal{S}_i} f(j)$.*

*Proof.* By Lemma 2.2, we have

$$\mathcal{L}_f(\mathcal{S}_i) = \sum_{j\in\mathcal{S}_i, k\in\bar{\mathcal{S}}_i} w_{jk}(f(j)-f(k)) \leq \sum_{j\in\mathcal{S}_i, j\in\bar{\mathcal{S}}_i} w_{jk}(f(u)-f(v))$$

$$\leq (f(u)-f(v)) \sum_{j\in\mathcal{S}_i, k\in\bar{\mathcal{S}}_i} w_{jk} = (f(u)-f(v))w(\mathcal{S}_i, \bar{\mathcal{S}}_i),$$

which concludes the proof. $\qquad\square$

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

This idea is based on two observations from our analysis: 1) the variance of the probabilities within each cluster is small; 2) there is a gap between the clusters. The small variance indicates that comparing the probabilities is essentially the same as comparing their means within clusters. The gap between the clusters ensures that the normalization makes the vectors align well. Lemma 3.3 makes this point precise. Our above analysis applies to multi-class problems and allows more than one labeled points in one class. In this general case, the classification rule is as follows: 1) compute the absorption probability vector $f_i : \mathcal{U} \to \mathbb{R}$ for each labeled point $i$ by taking all other labeled points as negative, where $\mathcal{U}$ denotes the set of unlabeled points; 2) normalize $f_i$ by its mass, denoted by $f_i'$; 3) assign each unlabeled point $j$ to the class of $j^* := \arg\max_i\{f_i'(j)\}$. We denote this algorithm as ARW-N-1NN.

**Lemma 3.3.** *Let $f_1 : \mathcal{V} \to \mathbb{R}$, $f_2 : \mathcal{V} \to \mathbb{R}$ be non-negative, and let $f_1' : \mathcal{V} \to \mathbb{R}$, $f_2' : \mathcal{V} \to \mathbb{R}$, where $f_1'(i) = f_1(i)/\sum_j f_1(j)$, $f_2'(i) = f_2(i)/\sum_j f_2(j)$. Suppose $\{C_1, C_2\}$ is a 2-partitioning of $\mathcal{V}$, i.e., $C_1 \cup C_2 = \mathcal{V}$, $C_1 \cap C_2 = \emptyset$, and denote $u_1 = \frac{1}{|C_1|}\sum_{i \in C_1} f_1(i)$, $v_1 = \frac{1}{|C_2|}\sum_{i \in C_2} f_1(i)$, $u_2 = \frac{1}{|C_1|}\sum_{i \in C_1} f_2(i)$, $v_2 = \frac{1}{|C_2|}\sum_{i \in C_2} f_2(i)$, $u_1' = \frac{1}{|C_1|}\sum_{i \in C_1} f_1'(i)$, $v_1' = \frac{1}{|C_2|}\sum_{i \in C_2} f_1'(i)$, $u_2' = \frac{1}{|C_1|}\sum_{i \in C_1} f_2'(i)$, $v_2' = \frac{1}{|C_2|}\sum_{i \in C_2} f_2'(i)$. If $u_1 > v_1$ and $u_2 < v_2$, then $u_1' > u_2'$ and $v_1' < v_2'$.*

*Proof.* Note that we have $u_1' > v_1'$ and $u_2' < v_2'$ since $u_1 > v_1$ and $u_2 < v_2$. Assume, to the contrary, that $u_1' \le u_2'$. Then $v_2' > u_2' \ge u_1' > v_1'$. Thus $1 = u_1'|C_1| + v_1'|C_2| < u_2'|C_1| + v_2'|C_2| = 1$, implying that the assumption is not correct. So we have $u_1' > u_2'$. The case for $v_1' < v_2'$ can be shown similarly. $\square$

## 3.2 Partially Absorbing Random Walks

Here we revisit the recently proposed partially absorbing random walks (PARW) [24], which generalizes absorbing random walks by allowing partial absorption at each state. The absorption rate $p_{ii}$ at state $i$ is defined as $p_{ii} = \frac{\alpha\lambda_i}{\alpha\lambda_i + d_i}$, where $\alpha > 0$, $\lambda_i > 0$ are regularization parameters. Given current state $i$, a PARW in the next step will get absorbed at $i$ with probability $p_{ii}$ and with probability $(1 - p_{ii}) \times \frac{w_{ij}}{d_i}$ moves to state $j$. Let $a_{ij}$ be the probability that a PARW starting from state $i$ gets absorbed at state $j$ within finite steps, and denote by $A = [a_{ij}] \in \mathbb{R}^{n \times n}$ the absorption probability matrix. Then $A = (\alpha\Lambda + L)^{-1}\alpha\Lambda$, where $\Lambda = \mathrm{diag}(\lambda_1, \ldots, \lambda_n)$ is the regularization matrix.

PARW is a unified framework with several popular SSL methods and PageRank [17] as its special cases, corresponding to different $\Lambda$. Particularly, the case $\Lambda = I$ has been justified in capturing the cluster structures [24]. In what follows, we extend this result to show that the columns of $A$ obtained by PARW with almost arbitrary $\Lambda$ (not just $\Lambda = I$) actually exhibit strong harmonic structures and should be expected to work equally well.

Our first observation is that while $A$ is not symmetric for arbitrary $\Lambda$, $A\Lambda^{-1} = (\alpha\Lambda + L)^{-1}\alpha$ is.

**Lemma 3.4.** $a_{ij} = \frac{\lambda_j}{\lambda_i}a_{ji}$.

*Proof.* $A\Lambda^{-1} = (\alpha\Lambda + L)^{-1}\alpha$ is symmetric. So $\frac{a_{ij}}{\lambda_j} = (A\Lambda^{-1})_{ij} = (A\Lambda^{-1})_{ji} = \frac{a_{ji}}{\lambda_i}$. $\square$

**Lemma 3.5.** *$a_{ii}$ is the only largest entry in the $i$-th column of $A$, $i = 1, \ldots, n$.*

*Proof.* Since $Q = (q_{ij}) := A\Lambda^{-1} = (\alpha\Lambda + L)^{-1}\alpha$ is symmetric and each column of $A$ is the corresponding column in $Q$ multiplied by a positive scale, it suffices to prove that $q_{ii}$ is the largest in the row $i$ of $Q$.

Note that $B = (b_{ij}) := \alpha\Lambda + D - W$ is symmetric and strictly diagonally dominant, i.e., $b_{kk} > \sum_{\ell \ne k}|b_{k\ell}|$ for any $k$. Assume, to the contrary, there exists $i, j, i \ne j$, such that $q_{ii} \le q_{ij}$. Denote $k = \arg\max_{\ell \ne i} q_{i\ell}$. Note that $Q > 0$ since $A > 0$. Then by $BQ = \alpha I$, we have $0 = B(k,:)Q(:,i) = \sum_\ell b_{k\ell}q_{\ell i} = b_{kk}q_{ki} + \sum_{\ell \ne k} b_{k\ell}q_{\ell i} \ge b_{kk}q_{ki} - \sum_{\ell \ne k}|b_{k\ell}|q_{ki} = (b_{kk} - \sum_{\ell \ne k}|b_{k\ell}|)q_{ki} > 0$. This contradicts the assumption, and thus completes the proof. $\square$

Our second observation is that the harmonic structure exists in the probabilities of PARW from every vertex getting absorbed at a particular vertex, i.e., in the columns of $A$. W.l.o.g., consider the first

column of $A$ and denote it by $p$. Assume that the vertices are sorted such that $p(1) > p(2) \geq \cdots \geq p(n-1) \geq p(n)$, where $p(1) > p(2)$ is due to Lemma 3.5. By the first step analysis of PARW, we can write $p$ in a recursive form:

$$p(1) = \frac{\alpha\lambda_1}{d_1 + \alpha\lambda_1} + \sum_{k\sim 1}\frac{w_{1k}}{d_1 + \alpha\lambda_1}p(k), \quad p(i) = \sum_{k\sim i}\frac{w_{ik}}{d_i + \alpha\lambda_i}p(k),\ i = 2,\ldots,n, \qquad (6)$$

which is equivalent to the following harmonic form:

$$p(1) = \frac{\alpha\lambda_1}{d_1}(1 - p(1)) + \sum_{k\sim 1}\frac{w_{1k}}{d_1}p(k), \quad p(i) = -\frac{\alpha\lambda_i}{d_i}p(i) + \sum_{k\sim i}\frac{w_{ik}}{d_i}p(k),\ i = 2,\ldots,n. \qquad (7)$$

The harmonic loss of $p$ can be computed from Eq. (7).

**Corollary 3.6.** $\mathcal{L}_p(\mathcal{S}_i) = \alpha\lambda_1(1 - \sum_{k\in\mathcal{S}_i}a_{1k}) = \alpha\lambda_1\sum_{k\in\bar{\mathcal{S}}_i}a_{1k},\ i = 1,\ldots,n-1$.

*Proof.* By Definition 2.1 and Eq. (7), we have

$$\mathcal{L}_p(\mathcal{S}_i) = \sum_{k=1}^{i}\mathcal{L}_p(k) = \alpha\lambda_1(1 - p(1)) - \sum_{k=2}^{i}\alpha\lambda_k p(k)$$

$$= \alpha\lambda_1 - \alpha\lambda_1 a_{11} - \sum_{k=2}^{i}\alpha\lambda_k a_{k1} = \alpha\lambda_1 - \alpha\lambda_1 a_{11} - \sum_{k=2}^{i}\alpha\lambda_1 a_{1k} = \alpha\lambda_1\sum_{k\in\bar{\mathcal{S}}_i}a_{1k},$$

where we have used Lemma 3.4 and the fact that $\sum_k a_{1k} = 1$. □

**Corollary 3.7.** *$p$ is left-continuous.*

*Proof.* This follows from $\mathcal{L}_p(i) = -\alpha\lambda_i p(i) < 0$. □

Now we are ready to examine the variation of $p$. Note that $\sum_k a_{1k} = 1$ and $a_{1k} \to \lambda_k/\sum_i\lambda_i$ as $\alpha \to 0$ [24]. By Theorem 2.7, the drop of $p(i)$ is upper bounded by $\alpha\lambda_1/w(\mathcal{S}_i,\bar{\mathcal{S}}_i)$, which is small when the cut $w(\mathcal{S}_i,\bar{\mathcal{S}}_i)$ is dense and $\alpha$ is small. Now let $k$ be the largest number such that $d(\mathcal{S}_k) \leq \frac{1}{2}d(\mathcal{V})$, and assume $\sum_{i\in\bar{\mathcal{S}}_k}\lambda_i \geq \frac{1}{2}\sum_i\lambda_i$. By Theorem 2.8, for $1 \leq i \leq k$, the drop of $p(i)$ across the cut $\{\mathcal{S}_i,\bar{\mathcal{S}}_i\}$ is lower bounded by $\frac{1}{3}\alpha\lambda_1/w(\mathcal{S}_i,\bar{\mathcal{S}}_i)$, if $\alpha$ is sufficiently small. This shows that $p(i)$ will drop a lot when the cut $w(\mathcal{S}_i,\bar{\mathcal{S}}_i)$ is weak. The comparison between the corresponding row and column of $A$ is shown in Figs. 3(a–b)[2], which confirms our analysis.

It is worth mentioning that our analysis substantially extends the results in [24] by showing that the setting of $\Lambda$ is not really necessary – a random $\Lambda$ can perform equally well using the columns instead of the rows of $A$. In addition, our result includes the seminal local clustering model [2] as a special case, which corresponds to $\Lambda = D$ in our analysis.

### 3.3 Pseudo-inverse of the Graph Laplacian

The pseudo-inverse $L^\dagger$ of the graph Laplacian is a valid kernel corresponding to commute times [10, 12]. While commute times may fail to capture the global topology in large graphs [22], $L^\dagger$, if used directly as a similarity measure, gives superior performance in practice [10]. Here we provide a formal analysis and justification for $L^\dagger$ by revealing the strong harmonic structure hidden in it.

**Lemma 3.8.** $(L^\dagger L)_{ij} = -\frac{1}{n},\ i \neq j$; and $(L^\dagger L)_{ii} = 1 - \frac{1}{n}$.

*Proof.* Denote by $L = U\Lambda U^\top$ the eigen-decomposition of $L$. Then $L^\dagger L = U\Lambda^\dagger\Lambda U^\top = I - \frac{1}{n}\mathbf{1}\mathbf{1}^\top$, where we have used the fact that $\mathbf{1}/\sqrt{n}$ is the eigenvector of $L$ associated with eigenvalue zero. □

Note that $L^\dagger$ is symmetric since $L$ is symmetric. W.l.o.g., we consider the first row of $L^\dagger$ and denote it by $\ell$. The following lemma shows the harmonic form of $\ell$.

Figure 3: (a) Absorption probabilities that a PAWR gets absorbed at other points when starting from $i$ (see Fig. 2). (b) Absorption probabilities that PAWR gets absorbed at $i$ when starting from other points. (c) The $i$-th row of $L^\dagger$. (d) Hitting times from $i$ to hit other points. (e) Hitting times from other points to hit $i$. (f) and (g) Eigenvectors of $L$ ($\min_i\{d_i\} = 0.0173$). (h) An eigenvector of $L_{sym}$. (i) and (j) Eigenvectors of $L_{rw}$. The values in (f–j) denote eigenvalues.

**Lemma 3.9.** $\ell$ *has the following harmonic form:*

$$\ell(1) = \frac{1-\frac{1}{n}}{d_1} + \sum_{k\sim 1}\frac{w_{1k}}{d_1}\ell(k), \ \ell(i) = -\frac{\frac{1}{n}}{d_i} + \sum_{k\sim i}\frac{w_{ik}}{d_i}\ell(k), \ i = 2,\ldots,n. \tag{8}$$

*Proof.* By Lemma 3.8, we have

$$1 - \frac{1}{n} = (L^\dagger L)_{11} = L^\dagger(1,:)L(:,1) = \sum_k \ell(k)L(k,1) = \ell(1)d_1 - \sum_{k\sim 1}\ell(k)w_{k1}, \tag{9}$$

giving $\ell(1) = \frac{1-\frac{1}{n}}{d_1} + \sum_{k\sim 1}\frac{w_{1k}}{d_1}\ell(k)$ using symmetry $w_{k1} = w_{1k}$.

Similarly, for $i > 1$, we have

$$-\frac{1}{n} = (L^\dagger L)_{1i} = L^\dagger(1,:)L(:,i) = \sum_k \ell(k)L(k,i) = \ell(i)d_i - \sum_{k\sim i}\ell(k)w_{ki}, \tag{10}$$

which yields $\ell(i) = -\frac{\frac{1}{n}}{d_i} + \sum_{k\sim i}\frac{w_{ik}}{d_i}\ell(k)$ using symmetry $w_{ki} = w_{ik}$. $\square$

W.l.o.g., assume the vertices have been sorted such that $\ell(1) > \ell(2) \geq \cdots \geq \ell(n-1) \geq \ell(n)$[3]. Then the harmonic loss of $\ell$ on the set $\mathcal{S}_i$ admits a very simple form, as shown below.

**Corollary 3.10.** $\mathcal{L}_\ell(\mathcal{S}_i) = \frac{|\bar{\mathcal{S}}_i|}{n}$, $i = 1,\ldots,n-1$.

*Proof.* By Definition 2.1, we have

$$\mathcal{L}_\ell(\mathcal{S}_i) = \sum_{k=1}^i d_k\left(\ell(k) - \sum_{j\sim k}\frac{w_{kj}}{d_k}\ell(j)\right) = d_1\left(\frac{1-\frac{1}{n}}{d_1}\right) + \sum_{k=2}^i d_k\left(-\frac{\frac{1}{n}}{d_k}\right) = \frac{|\bar{\mathcal{S}}_i|}{n}. \tag{11}$$

$\square$

**Corollary 3.11.** $\ell$ *is left-continuous.*

*Proof.* By Lemma 3.9, the harmonic loss for each vertex $i = 2,\ldots,n$ is negative (i.e., $-\frac{1}{n}$). Thus $\ell$ is left-continuous according to Proposition 2.6. $\square$

By Corollary 3.10, $\mathcal{L}_\ell(\mathcal{S}_i) < 1$ and decreases very slowly in large graphs since $\mathcal{L}_\ell(\mathcal{S}_i) - \mathcal{L}_\ell(\mathcal{S}_{i+1}) = \frac{1}{n}$ for any $i$. From the analysis in Sec. 2, we can immediately conclude that the variation of $\ell(i)$ is dominated by the cut cost on the superlevel set $\mathcal{S}_i$. Fig. 3(c) illustrates this argument.

### 3.4 Hitting Times

The hitting time $h_{ij}$ from vertex $i$ to $j$ is the expected number of steps it takes a random walk starting from $i$ to reach $j$ for the first time. While it was proven in [22] that hitting times are dominated by the local structure of the target, we show below that the hitting times from other points to the same target admit a harmonic structure, and thus are still able to capture the global structure of graphs. Our result is complementary to the analysis in [22], and provides a justification of using hitting times in information retrieval where the query is taken as the target to be hit by others [15].

Let $h : \mathcal{V} \to \mathbb{R}$ be the hitting times from every vertex to a particular vertex. W.l.o.g., assume the vertices have been sorted such that $h(1) \geq h(2) \geq \cdots \geq h(n-1) > h(n) = 0$, where vertex $n$ is the target vertex. Applying the first step analysis, we obtain the harmonic form of $h$:

$$h(i) = 1 + \sum_{k \sim i} \frac{w_{ik}}{d_i} h(k), \quad \text{for } i = 1, \ldots, n-1. \tag{12}$$

The harmonic loss on the set $\mathcal{S}_i$ turns out to be the volume of the set, as stated below.

**Corollary 3.12.** $\mathcal{L}_h(\mathcal{S}_i) = \sum_{1 \leq k \leq i} d_k$, $i = 1, \ldots, n-1$.

*Proof.* By Definition 2.1, we have

$$\mathcal{L}_h(\mathcal{S}_i) = \sum_{k=1}^{i} d_k \left( h(k) - \sum_{j \sim k} \frac{w_{kj}}{d_k} h(j) \right) = \sum_{1 \leq k \leq i} d_k.$$

$\square$

**Corollary 3.13.** $h$ *is right-continuous.*

*Proof.* By Eq. (12), the harmonic loss at each vertex is positive. Thus $h$ is right-continuous according to Proposition 2.6. $\square$

Now let us examine the variation of $h$ across any cut $\{\mathcal{S}_i, \bar{\mathcal{S}}_i\}$. Note that

$$\frac{\mathcal{L}_h(\mathcal{S}_i)}{w(\mathcal{S}_i, \bar{\mathcal{S}}_i)} = \frac{\alpha_i}{\Phi(\mathcal{S}_i)}, \quad \text{where } \alpha_i = \frac{d(\mathcal{S}_i)}{\min(d(\mathcal{S}_i), d(\bar{\mathcal{S}}_i))}. \tag{13}$$

First, by Theorem 2.8, there could be a significant gap between the target and its neighbors, since $\alpha_{n-1} = \frac{d(\mathcal{V})}{d_n} - 1$ could be quite large. As $i$ decreases from $d(\mathcal{S}_i) > \frac{1}{2}d(\mathcal{V})$, the variation of $\alpha_i$ becomes slower and slower ($\alpha_i = 1$ when $d(\mathcal{S}_i) \leq \frac{1}{2}d(\mathcal{V})$), so the variation of $h$ will depend on the variation of the conductance of $\mathcal{S}_i$, i.e., $\Phi(\mathcal{S}_i)$, according to Theorems 2.7 and 2.8. Fig. 3(e) shows that $h$ is flat within the clusters, but there is a large gap presented between them. In contrast, there are no gaps exhibited in the hitting times from the target to other vertices (Fig. 3(d)).

### 3.5 Eigenvectors of the Laplacian Matrices

The eigenvectors of the Laplacian matrices play a key role in graph partitioning [20]. In practice, the eigenvectors with smaller (positive) eigenvalues are more desired than those with larger eigenvalues, and the ones from a normalized Laplacian are preferred than those from the un-normalized one. These choices are usually justified from the relaxation of the normalized cuts [18] and ratio cuts [11]. However, it has been known that these relaxations can be arbitrarily loose [20]. It seems more interesting if one can draw conclusions by analyzing the eigenvectors directly. Here we address these issues by examining the harmonic structures in these eigenvectors.

We follow the notations in [20] to denote two normalized graph Laplacians: $L_{rw} := D^{-1}L$ and $L_{sym} := D^{-\frac{1}{2}}LD^{-\frac{1}{2}}$. Denote by $u$ and $v$ two eigenvectors of $L$ and $L_{rw}$ with eigenvalues $\lambda_u > 0$ and $\lambda_v > 0$, respectively, i.e., $Lu = \lambda_u u$ and $L_{rw}v = \lambda_v v$. Then we have

$$u(i) = \sum_{k \sim i} \frac{w_{ik}}{d_i - \lambda_u} u(k), \quad v(i) = \sum_{k \sim i} \frac{w_{ik}}{d_i(1 - \lambda_v)} v(k), \quad \text{for } i = 1, \ldots, n. \tag{14}$$

We can see that the smaller $\lambda_u$ and $\lambda_v$, the stronger the harmonic structures of $u$ and $v$. This explains why in practice the eigenvector with the second[4] smallest eigenvalues gives superior performance. As long as $\lambda_u \ll \min_i\{d_i\}$, we are safe to say that $u$ will have a significant harmonic structure, and thus will be informative for clustering. However, if $\lambda_u$ is close to $\min_i\{d_i\}$, no matter how small $\lambda_u$ is, the harmonic structure of $u$ will be weaker, and thus $u$ is less useful. In contrast, from Eq. (14), $v$ will always enjoy a significant harmonic structure as long as $\lambda_v$ is much smaller than 1. This explains why eigenvectors of $L_{rw}$ are preferred than those of $L$ for clustering. These arguments are validated in Figs. 3(f–j), where we also include an eigenvector of $L_{sym}$ for comparison.

## 4   Experiments

In the first experiment, we test absorbing random walks (ARW) for SSL, with the class mass normalization suggested in [27] (ARW-CMN), our proposed normalization (ARW-N-1NN, Sec. 3.1), and without any normalization (ARW-1NN) – where each unlabeled instance is assigned the class of the labeled instance at which it most likely gets absorbed. We also compare with the local and global consistency (LGC) method [25] and the PARW with $\Lambda = I$ in [24], where the regularization parameters are set as 0.9 and $1e - 6$, respectively. We use 9 real data sets for this experiment, including USPS, YaleB, and 7 frequently used UCI datasets, as summarized in Table 1.

We construct a weighted 20-NN graph for each data set, except for the YaleB, imageseg, and iris data sets, where we build 50-NN, 50-NN, and 25-NN graphs respectively to ensure the graphs are connected. The similarity between vertices $i$ and $j$ is computed as $w_{ij} = \exp(-d_{ij}^2/\sigma)$ if $i$ is within $j$'s $k$ nearest neighbors or vice versa, and $w_{ij} = 0$ otherwise, where $\sigma = 0.2 \times r$ with $r$ as the average square distance between each point to its 20th nearest neighbor. For USPS and YaleB, we randomly sample 20 instances as labeled data; while for others, we randomly sample 10 instances. The sampling process makes sure at least one label is sampled for each class. Each classification accuracy is averaged over 100 trials.

The results are summarized in Table 2. We can see that ARW-N-1NN and PARW ($\Lambda = I$) consistently perform the best, which verifies our analysis in Sec. 3. The results of ARW-1NN are unsatisfactory due to its bias to the labeled instance with the largest degree [1]. While ARW-CMN does improve over ARW-1NN in many cases, it does not perform as well as ARW-N-1NN, mainly because of the artifacts induced by estimating the class proportion from limited labeled data. The results of LGC are not comparable to ARW-N-1NN and PARW ($\Lambda = I$), which is probably due to the lack of a harmonic structure[5].

Table 1: The 9 datasets tested in the experiments.

|  | USPS | YaleB | satimage | imageseg | ionosphere | iris | protein | spiral | soybean |
|---|---|---|---|---|---|---|---|---|---|
| # examples | 9298 | 5760 | 6435 | 2310 | 351 | 150 | 116 | 100 | 47 |
| # classes | 10 | 10 | 6 | 7 | 2 | 3 | 6 | 2 | 4 |
| # dimensions | 256 | 1200 | 36 | 19 | 34 | 4 | 20 | 3 | 35 |

Table 2: Classification accuracy on 9 datasets.

|  | USPS | YaleB | satimage | imageseg | ionosphere | iris | protein | spiral | soybean |
|---|---|---|---|---|---|---|---|---|---|
| ARW-N-1NN | **.879** | **.892** | **.777** | **.673** | **.771** | **.918** | **.589** | **.830** | **.916** |
| ARW-1NN | .445 | .733 | .650 | .595 | .699 | .902 | .440 | .754 | .889 |
| ARW-CMN | .775 | .847 | .741 | .624 | .724 | .894 | .511 | .726 | .856 |
| LGC | .821 | .884 | .725 | .638 | .731 | .903 | .477 | .729 | .816 |
| PARW ($\Lambda = I$) | .880 | .906 | .781 | .665 | .752 | .928 | .572 | .835 | .905 |

In the second experiment, we test PARW on a retrieval task on the whole USPS dataset (Table 1). We compare the cases with $\Lambda = I$ and $\Lambda = R$, where $R$ is a random diagonal matrix with positive diagonal entries sampled from the uniform distribution on the interval $[0, 1]$. For $\Lambda = R$, we also compare the uses of columns and rows for retrieval. Following [24], we set $\alpha = 1e - 6$, use each instance in USPS as a query for the entire dataset, and report the mean average precision (MAP). The graph construction is the same as in the first experiment.