[Reviews · NeurIPS 2013]

Submitted by Assigned_Reviewer_2

This paper proposes a harmonic loss-based analysis method to measure the agreement between a target function and the underlying graph. The authors apply this analysis to many commonly used graph-based learning methods with interesting insights. The paper is well written and the analysis should be of interest to the general NIPS community.

Graph-based SSL and random walks and their connections to harmonic functions are well recognized in previous research, which makes the results in the paper, although novel, somewhat incremental.

There are a few typos in the paper which I assume the authors will be able find with a careful reading.
Summary: This paper proposes a harmonic loss-based analysis method to measure the agreement between a target function and the underlying graph. The paper is well written and the analysis should be of interest to the general NIPS community.

Submitted by Assigned_Reviewer_5

- A desirable property (*) for algorithms on graphs is to get a solution which is almost constant in each cluster, but drastically change between clusters.
This paper tries to establish a relation between ``near harmonic'' function on graphs and their drop in dense/sparse areas. The authors then study different algorithms and their ``harmonic loss'', as a step to show that property (*) holds for their results.
- This paper contains several mini steps toward getting tools to check if the property (*) holds for a machine algorithm. However the results are not still strong enough, and can not be considered as a proof.
- The paper is easy to read and follow, but there are some space for improvements. Some of the results can significantly simplified (See detailed comments). Supplementary material is supposed to contain proofs and specific details, and not complete new sections (Section 4.2 and 4.3)
- Originality and significance: The paper introduce new tools to intuitively justify the property (*) for graph algorithms. The machinery looks attractive for me, but i do not see them as a tool for finding rigorous proofs.

Main comments:
- The so called left/right/- continuity of functions on graphs as it is defined, have nothing to do with mathematical concept of continuity. It does not guaranty any bound on function's jump. If you remove all results about continuity, would the message of the paper is change? My suggestion is to choose another keyword, which does not imply properties stronger than the actual meaning to the reader.
- lines 192-199: All the statements are imprecise and hand wavy. One can build functions f which is almost harmonic everywhere, but does not drop at all on a cut with a very small w(). Consider this graph and function values defined on vertices, where 000 shows a big complete graph with f zero on it (or super tiny random values if you like):
1--0.8--0.6--0.4--0.2--0--000--0--0--000
f is almost harmonic everywhere(except two vertices), however it does not drop at all on the edges between two 000 clusters, where conductance is relatively small.
For other statements in this paragraph, also one can build counterexamples.
- In Corollary 3.1, we can explicitly characterize L_f(S_i), which is the inverse of the resistance distance between vertex 1 and n
- line 247: it is claimed that f continuously changes on graphs. However the definition 2.5 of continuous f is much weaker than "continuously changing on graphs"
- lines 264-310: We have f_1(i) + f_2(i) = 1. So we do not need to consider f_1 and f_2 as one of them already contain all the information. Also Lemma 3.3 boils down to deciding the class by comparing f'_1(i) with 1/n.
- I suggest the Authors to give a recipe for analyzing semi-supervised algorithms. This would increase the readability of their paper. A recipe is something like:
1- try to write the solution function f as a sum of a harmonic term and non-harmonic term. In my own words, find the net incoming flow into each node. It is clear that we may not be able to write the solution for all methods in this form.
2- Use theorems 2.7 and 2.8 to check if they provide good bounds on the drop
- There are many qualitative statements, which Authors should avoid having them. This makes their work look very hand-wavy and imprecise.



tiny comments:
- In definition 2.5, it is good to mention that S_i depends on f.
- line 170: r.h of inequality, k \in \bar{S_i}

Pro:
- Bound on the change of function f depending on Harmonic Loss and conductance
- Presenting the solution of different graph algorithms in the harmonic form

Co:
- Upper/lower bounds on the change of function f only indicate the property (*) and does not give a rigorous proof
- The normalizing step can presented much simpler. In fact i do not count it as a contribution of the paper.
- The supplementary material is misused by Authors.
Summary: This paper presents new bounds on the change of function f depending on ``Harmonic Loss'' and conductance. Also the solution of different graph algorithms are presented in a harmonic form. The results are implications for the property (*), but should not considered as a rigorous proof.

There are some novelties and interesting ideas hidden in the paper, but the Authors did not explore their ideas well enough.

Submitted by Assigned_Reviewer_8

The paper studied harmonic functions in several graph Laplacian based semi-supervised learning algorithms. The key finding is that “the associate target function can faithfully capture the graph topology in that it drops little in dense area and sharply otherwise.”

As mentioned in the paper, graph-based semi-supervised learning has been studied from many aspects. Mostly they are either random walk based or Laplacian based algorithms. This paper tried to unify different studies under one framework, which is a nice motivation.

However, the role of harmonic functions has been studied well enough in this area I think. Random walks on regular lattice and diffusion equations are closely related to each other. In fact, with certain mild conditions, continuous diffusion process is the limit of random walks as the lattice become denser and denser. Moreover, diffusion process is described by Laplace equation. All of these can be found in most basic random walk text books. Then, the next piece of the puzzle is weighted graph and its limit, manifold, and graph Laplacian and Laplace-Beltrami operator, which were solved by the PhD dissertations of Dr. Mikhail Belkin and Dr. Stéphane Lafon. A recent new puzzle [11] was also solved by [20]. All of these works are already very clear about the role of harmonic functions in these algorithms.

I think the research of this area should be way past the random walk intuition analysis stage, which was great initially. Ultimately, it is a function estimation problem.

Supplementary material should be the content different from the main paper, not the whole paper.

Overall, the paper is nicely written. However, the content is not significant or new enough for publication compared to other submissions.
Summary: This paper tried to unify different random walk based or Laplacian based semi-supervised learning algorithms under one framework, which is a nice motivation. However, most of the study in this paper is not new or significant enough for the conference compared to other submissions.

Submitted by Assigned_Reviewer_9

The authors introduce a functional called the "harmonic loss" and show that (a) it characterizes smoothness in the sense that functions with small harmonic loss change little across large cuts (to be precise, the cut has to be a level set separator) (b) several algorithms for learning on graphs implicitly try to find functions that minimize the harmonic loss, subject to some constraints.

The "harmonic loss" they define is essentially the (signed) divergence \nabla f of the function across the cut, so it's not surprising that it should be closely related to smoothness. In classical vector calculus one would take the inner product of this divergence with itself and use the identity

< \nabla f, \nabla f > = < f, \nabla^2 f >

to argue that functions with small variation, i.e., small | \nabla f |^2 almost everywhere can be found by solving the Laplace equation. On graphs, modulo some tweaking with edge weights, essentially the same holds, leading to minimizing the quadratic form f^\top L f, which is at the heart of all spectral methods. So in this sense, I am not surprised.

Alternatively, one can minimize the integral of | \nabla f |, which is the total variation, and leads to a different type of regularization (l1 rather than l2 is one way to put it). The "harmonic loss" introduced in this paper is essentially this total variation, except there is no absolute value sign. Among all this fairly standard stuff, the interesting thing about the paper is that for the purpose of analyzing algorithms one can get away with only considering this divergence across cuts that separate level sets of f, and in that case all the gradients point in the same direction so one can drop the absolute value sign. This is nice because the "harmonic loss" becomes linear and a bunch of things about it are very easy to prove. At least this is my interpretation of what the paper is about.

I would have preferred to see an exposition that explicitly makes this connection to divergence, total variation, etc.. That would also have made it clear why the level sets are interesting.

In general, seeing a unifying analysis of different algorithms for learning on graphs is interesting and enlightening. I can imagine some of the results from the paper making their way into textbooks eventually. Although, again, if the connection to classic notions had been made clear, the article would be even more "unifying". I am sure that somebody somewhere has looked at notions of divergence and total variation on graphs, just not necessarily in the context of machine learning algorithms. Learning on graphs is an important topic, lots of different algorithms have been proposed, and any paper that clears up the dust and finds the common underlying mathematics is potentially useful.

On the other hand, this paper doesn't propose a new algorithm, and the "corrections [to exisiting algorithms] obtained from the new insights" are relatively minor. Correspondingly, the experiments are not so remarkable.
Summary: I am in two minds about this paper. On the one hand, I think that the community should learn about these connections between the different algorithms for learning on graphs. On the other hand the "harmonic loss" that the authors introduce is not very surprising and could be better related to standard concepts in math such as divergence, total variation, etc., which work in both discrete and the continuous domain.
Author Feedback

Author rebuttal: We thank all the reviewers for their comments.


To Reviewer_5

Q1. Upper/lower bound on the change of function f does not give a rigorous proof of property (*).

A1. Note that we take a general approach without assuming any special structure about graphs. This ensures the derived bounds are applicable to any function on any graph, and makes it easy to analyze property (*) with the bounds. If more assumptions are incorporated, we expect it would be quite feasible to derive additional proofs for certain functions, which we consider as an interesting line of future work.

Q2. The normalizing step can’t be counted as a contribution of the paper.

A2. The normalization scheme is suggested by our analysis and most importantly it leads to significant performance gains over prior works.

Q3. Supplementary material is supposed to contain proofs and specific details, and not complete new sections (Sec 4.2 and 4.3)

A3. Thanks much for the reminder. We will do so in the final version. But we’d like to mention that Sec 4.2 and 4.3 are only intended to add more experimental evidence to confirm our analysis. They can be dropped without degrading the integrity of the paper.

Q4. Would removing all results about continuity change the message of the paper?

A4. Yes. The proposed notion of "continuity", though not as strong as the usual one in Math, is a desired property for functions on graphs. For example, a “discontinuous” function that changes alternatively among different clusters is considered bad.

Q5. The function 1--0.8--0.6--0.4--0.2--0--000--0--0—000 is almost harmonic everywhere (except two vertices), however it does not drop at all on the edges between two 000 clusters, where conductance is relatively small.

A5. 0--000--0--0—000 is a constant function. Our statement doesn’t include the case of constant functions. Because their harmonic loss is zero everywhere, by Theorems 2.7 & 2.8, they don’t drop at all.

Q6. Lemma 3.3 boils down to deciding the class by comparing f'_1(i) with 1/n.

A6. True, but this only holds for binary classification, while our proof of Lemma 3.3 generalizes to multi-class classification, as remarked in lines 236-241 (paper).


To Reviewer_8

Q7. The paper studied harmonic functions in several graph Laplacian based semi-supervised learning (SSL) algorithms. The role of harmonic functions has been studied well enough.

A7. There may be some misunderstanding here. This paper is not about harmonic functions, but any functions on graphs. It is not on SSL alone, but on graph-based learning in general, including classification, retrieval, and clustering.

First, the problem studied in this paper is fundamentally different from those in Dr. Belkin and Dr. Lafon’s theses, where they address the selection of the canonical basis on graphs and establish the asymptotic convergence on manifolds. Here we examine how functions on graphs deviate from being harmonic and develop bounds to analyze their theoretical behavior, which to the best of our knowledge, has not been explored before.

Second, our studies go much beyond SSL. In fact, 4 out of 5 models we investigate are unsupervised. For examples, our results justify the role of pseudo-inverse of graph Laplacian as a similarity measure for recommendation, and explain the choice of eigenvectors for spectral clustering.

Q8. A recent new puzzle [11] was also solved by [20].

A8. [11, 20] argue that first-order Laplacian regularization is not sufficient for high-dimensional problems and suggest using high-order regularization, however how to choose the order is unknown. In contrast, we show that first-order Laplacian regularization actually carries sufficient information for classification, which can be extracted with a simple normalization (lines 229-235). Our point is validated by extensive experiments with high-dimensional datasets.

Q9. Graph-based SSL should be way past the random walk intuition analysis stage.

A9. First, as mentioned in A7, this paper is not limited to random walk methods or SSL applications. Second, while various methods (random walk based or not) have been proposed for graph-based SSL learning, many of them are not well understood or even misused, as pointed out recently in [11,17,18]. Here is a quote from [17] (Luxburg et al. 2010), "In general, we believe that one has to be particularly careful when using random walk based methods for extracting global properties of graphs in order to avoid getting lost and converging to meaningless results."


To Reviewer_9

Thanks for pointing out the potential connection between “harmonic loss” and standard divergence and total variation, which seems to be an interesting direction. We expect such investigation would help make the concepts clearer, or more importantly, lead to more systematic tools for analyzing graph-based algorithms.